# Assessment of laboratory $O_4$ absorption cross-sections at 360 nm using atmospheric long-path DOAS observations

Bianca Lauster[1,2], Udo Frieß[2], Jan-Marcus Nasse[2,3], Ulrich Platt[2], and Thomas Wagner[1,2]

[1]Satellite Remote Sensing Group, Max Planck Institute for Chemistry, Mainz, Germany
[2]Institute for Environmental Physics, University of Heidelberg, Heidelberg, Germany
[3]Now at: Energie Baden-Württemberg AG, Karlsruhe, Germany

**Correspondence:** Bianca Lauster (b.lauster@mpic.de)

**Abstract.** Absorption of light in the atmosphere by collision-induced absorption by two oxygen molecules $O_2$-$O_2$, in the following referred to as $O_4$, can be used to derive properties of aerosols and clouds from remote sensing observations. In recent years, inconsistencies between the measured atmospheric $O_4$ absorption and radiative transfer simulations were found for Multi-AXis Differential Optical Absorption Spectroscopy (MAX-DOAS) measurements. In the presented study, over two years of observations from a long-path (LP-) DOAS instrument deployed at the German research station Neumayer, Antarctica, are analysed. While MAX-DOAS instruments measure spectra of scattered sunlight at different elevation angles, LP-DOAS utilises an artificial light source and the atmospheric absorptions are measured along a fixed (and well-defined) light path close to the surface. Further, the pristine measurement location allows to investigate the relation between measured and modelled $O_4$ absorption over a large range of temperatures (-45°C to +5°C). Overall good agreement is found between the retrieved $O_4$ absorption cross-sections covering the absorption band at 360 nm and laboratory measurements. While the best agreement is obtained for the Finkenzeller and Volkamer (2022) cross-sections, deviations at cold ambient temperatures (below ca. -25°C) are observed for the Thalman and Volkamer (2013) cross-sections. Other $O_4$ absorption bands could not be investigated because these are not (fully) within the spectral range of the measured spectra. This study strongly supports the accuracy of commonly used $O_4$ absorption cross-sections in DOAS analyses, while more work is needed to understand the earlier reported inconsistencies in MAX-DOAS observations.

## 1 Introduction

Absorption of light in the atmosphere by the oxygen collision complex $O_2$-$O_2$ (in the following referred to as $O_4$) in the UV-visible spectral range is commonly used in remote sensing applications to derive properties of aerosol and clouds. The collision of two oxygen molecules gives rise to the formation of several absorption bands through so-called collision-induced absorption (CIA). CIA describes the absorption in systems of interacting atoms or molecules where the absorption exceeds the simple sum of the absorption by the isolated atoms or molecules (Frommhold, 2006; Finkenzeller and Volkamer, 2022, and references therein). Since CIA does not involve a bound state, the term $O_4$ might be misleading. However, this term is rooted in the MAX-DOAS community for which reason the notation $O_4$ will be used in the following to denote $O_2$-$O_2$ CIA. The absorption by $O_4$ occurs proportionally to the square of the $O_2$ concentration. Deviations from the $O_4$ absorption against those

for clear sky conditions thus indicate changes in atmospheric radiative transfer which allows to retrieve properties of cloud and/or aerosol particles from such observations. This technique can be applied to remote sensing measurements of scattered sunlight performed by ground-based instruments, airborne platforms and satellites (e.g., Hönninger et al., 2004; Wagner et al., 2004, 2010; Wittrock et al., 2004; Frieß et al., 2006, 2016; Irie et al., 2008; Prados-Roman et al., 2011).

Over a decade ago, Wagner et al. (2009) first reported inconsistencies between the measured atmospheric $O_4$ absorption from Multi-AXis Differential Optical Absorption Spectroscopy (MAX-DOAS) observations and radiative transfer simulations. MAX-DOAS instruments measure scattered sunlight under various mostly slant elevation angles (Hönninger et al., 2004). To achieve agreement with the forward model simulations, Wagner et al. (2009) suggested applying a scaling factor (SF < 1) to the measured $O_4$ slant column densities (SCDs). Similar findings were then reported by, e.g., Clémer et al. (2010), Vlemmix et al. (2015), Frieß et al. (2016) and Wagner et al. (2021), who all found best agreement for SF between 0.75 and 0.9, while other studies (including direct sun measurements and aircraft measurements) did not see the necessity for a scaling factor (e.g., Spinei et al., 2015; Ortega et al., 2016).

A more detailed overview and discussion can be found in Wagner et al. (2019). Since there is still no consensus in the community on whether or not an SF is appropriate, this ambiguity leads to substantial uncertainties in the aerosol results derived from MAX-DOAS measurements.

In this study, long-term long-path (LP-) DOAS observations are used to examine the $O_4$ absorption at 360 nm. The spectra measured in the UV spectral range cover the wavelengths from about 327 to 395 nm, however, a shorter fit window from 352 to 387 nm was chosen here. Details on the data analysis are given in the next section.

The absorption band at 577 nm is not fully covered leading to a less stable retrieval and thus was excluded from a more detailed investigation. The different analyses include the commonly used absorption cross-sections of Thalman and Volkamer (2013) as well as Finkenzeller and Volkamer (2022) at different temperatures. Figure 1 shows the respective cross-sections at 293 K and the wavelength range covered by the data.

While both Thalman and Volkamer (2013) and Finkenzeller and Volkamer (2022) use a similar experimental set-up of a cavity enhanced (CE-) DOAS instrument, there are some improvements implemented for the more recent study optimising the optical components and thus the stability of the measurements. Moreover, pure oxygen was used instead of air mixtures to enhance the signal-to-noise considerably and allowing to capture also weak absorption features. Both studies report that the peak cross-section increases and the band width decreases at colder temperatures. However, Finkenzeller and Volkamer (2022) observe an increase of the integral cross-section with temperature which was not seen by Thalman and Volkamer (2013). This deviating observation is likely due to the assumption that the absorption cross-section is zero in the minima between neighbouring absorption bands to approximate the baseline of the absorption cross-section in Thalman and Volkamer (2013) which is not applied in Finkenzeller and Volkamer (2022). It should be noted that the Thalman and Volkamer (2013) absorption cross-sections are recommended in the current HITRAN database (Karman et al., 2019; Gordon et al., 2022).

At first, differences between MAX-DOAS and LP-DOAS measurements are introduced to answer the question why LP-DOAS observations are well suited to further investigate the reported inconsistencies from MAX-DOAS studies. After the following description of the measurement set-up and analysis, the main part focuses on the comparison between measured and

calculated $O_4$ absorptions. Lastly, the findings of this study are summarised and put into the context of previous and future work.

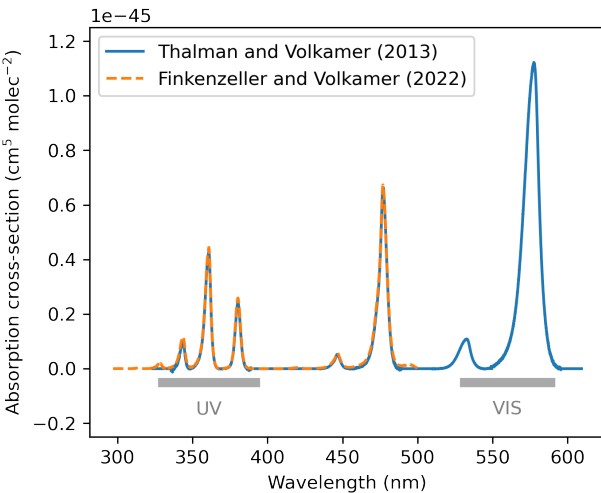

**Figure 1.** $O_4$ absorption cross-sections investigated in this study. The grey bars indicate the wavelength range measured by the LP-DOAS instrument. Note that the absorption band at 477 nm is not covered and the absorption band at 577 nm is only partly covered by the measurements. The figure depicts the absorption cross-sections at 293 K.

## 2 Long-path DOAS

### 2.1 Suitability for accurate atmospheric $O_4$ absorption measurements

Long-path (LP-) DOAS belongs to the active DOAS applications (Perner et al., 1976; Perner and Platt, 1980; Platt and Perner, 1983) and is a well-established remote sensing technique. In contrast to passive systems which measure scattered sunlight, such as MAX-DOAS instruments, LP-DOAS instruments use artificial light sources, for example a Xenon arc lamp or in more recent applications a laser driven light source. Thereby, continuous observations of trace gases are possible independent of natural light sources, i.e., also during night-time and in the deep UV. The most prominent advantage for the presented study is the well-defined light path of a LP-DOAS set-up. Along this light path, a mean trace gas concentration is determined. In MAX-DOAS analyses, the conversion from SCDs to concentrations always requires further processing steps including radiative transfer simulations and thus possibly leads to a higher uncertainty of the results. Especially at short wavelength, direct sun measurements, despite the well-defined light path (at high elevation angles), experience further difficulties such as small absorptions or substantial contributions of scattered sun light (at low elevation angles). Another benefit of LP-DOAS measurements is that the ambient temperature and pressure can be assumed constant along the given light path of a couple of kilometres at ground level, while this is not the case in MAX-DOAS retrievals or direct sun measurements, from which the atmospheric column density is derived considering vertical profiles of temperature and pressure. Nonetheless, the basis of the DOAS principle is applicable to both active and passive DOAS systems.

### 2.2 Instrument and measurement site

The LP-DOAS instrument used in this study was purpose-built for operation under polar conditions. Its observations complement the long-term measurements at the German research station Neumayer III, Antarctica, from January 2016 to August 2018. A detailed description of the instrument and its set-up can be found in Nasse (2019) and Nasse et al. (2019).

The LP-DOAS instrument couples light from a laser driven light source (EQ-99X) into a telescope located at the trace gas observatory of the Neumayer III station (Fig. 2) which creates a light beam that is transmitted through the atmosphere across a distance of 1.55 km to the closer reflector (Met retro), or to another retro-reflector at 2.95 km (Atka retro). Depending on the prevailing weather conditions, the amount of reflected light varies, and the light path can be chosen depending on the atmospheric conditions to optimise the amount of received light and the covered light path length. After reflection at one of the retro-reflector arrays, the light is received again by the same telescope doubling the length of the light path. Spectra are then captured by an Acton 300i spectrometer using a holographic grating (1200 gr. $mm^{-1}$) with attached Andor DU440 BU CCD. This set-up allows for a spectral resolution of ca. 0.54 nm covering a spectral window of about 65 nm. The measured spectra have a temporal resolution of about 2 to 30 minutes and are analysed without filtering for specific measurement conditions based on the DOAS principle. Details of the analysis are given in the next section.

The measurement site exhibits an extraordinarily large temperature range exceeding -35°C and +0°C as can be seen in Fig. 3. Apart from seasonal variations, these temperatures are driven by the advection of air masses of different origin, depending on

synoptic conditions. The annual cycle of surface pressure indicates the influence of large scale atmospheric patterns around the

Antarctic continent (König-Langlo et al., 1998).

Given its remote location, the aerosol optical depth is often below 0.1 (see, e.g., respective site on the AERONET webpage, NASA-GSFC) making the station an ideal location for this study. In particular, the majority of the measurements exhibit an only slightly attenuated signal after passing the atmospheric light path twice. This allows for long light paths and the concomitant strong $O_4$ absorption offers a good signal-to-noise ratio.

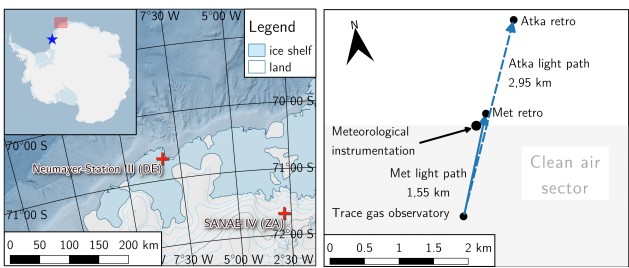

**Figure 2.** Location of the Neumayer III station on the Ekström ice shelf (left) and overview of the installations in the vicinity of the station (right). The closest neighbouring station is the South African base SANAE IV about 225 km to the south-east, while the British station Halley VI (blue star) is the only other Antarctic station located on an ice shelf and about 800 km south-west of Neumayer III (Schiermeier, 2004). Taken from Nasse (2019).

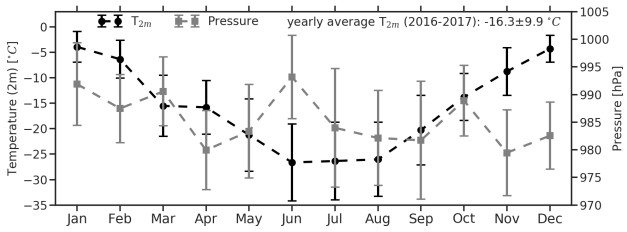

**Figure 3.** Monthly averages of 2 m temperature and atmospheric pressure from 2016 to 2017 at the Neumayer III station. For details, see Nasse (2019). The whiskers indicate one standard deviation of the data. Taken from Nasse (2019).

## 2.3 Data analysis

Average trace gas concentrations along the light path can be retrieved for each measured spectrum by making use of the Lambert-Beer law. For atmospheric applications, this is commonly referred to as the DOAS approach (Platt and Stutz, 2008). In the study presented here, the spectral analysis of the obtained spectra is performed in HeiDOAS (v1.2). This newly developed Python-based library was programmed for the analysis of DOAS measurements and conducts the fit using a Levenberg-Marquardt algorithm (Levenberg, 1944; Marquardt, 1963) allowing for the commonly used parameters in DOAS analyses.

While MAX-DOAS applications usually use zenith measurements as a reference, the LP-DOAS instrument can measure an absorption-free spectrum of the light source by moving a reference plate into the light path and thereby creating a short-cut for the light, which does not traverse the atmosphere but enters the spectrograph directly. To determine and correct the influence of the atmospheric background as well as instrumental contributions of the dark current signal from the CCD and the electronic offset, regular background spectra are recorded in addition without the light source. The wavelength calibration is accomplished by determining the channel-to-wavelength attribution from measurements of emission line spectra of a mercury and a neon vapour lamp. These calibration spectra were recorded regularly throughout the measurement campaign and also allow to retrieve the instrumental slit function needed for the convolution of the fitted absorption cross-sections.

The spectral resolution of the $O_4$ absorption cross-sections range from 0.32 to 0.45 nm and 0.31 to 0.42 nm for Thalman and Volkamer (2013) and Finkenzeller and Volkamer (2022), respectively. Since the observed absorption features are 10 to 30 times wider and also the spectral resolution of the LP-DOAS (0.54 nm) is lower, the absorption cross-sections can be considered as "fully resolved" and are directly convolved with the respective instrument slit function.

After the preprocessing of the data, i.e. conducting the wavelength calibration and accounting for the background, the measured spectra are fitted according the DOAS fit settings given in Table 1. In order to remove broad-band spectral features that are caused by the light source or the optical components and can have characteristic shapes, a binomial high-pass filter and a low order closure polynomial is applied. Different analyses will be shown in the following, including one of the $O_4$ absorption cross-sections listed in the table.

To account for quickly varying atmospheric conditions between the acquisition of the atmospheric spectrum and the respective background spectrum, an atmospheric background spectrum is also included in the DOAS fit which becomes important if the background correction cannot fully compensate for the background signal (Pöhler, 2010).

Figure 4 exemplarily shows a fit in the UV spectral range with the clearly visible $O_4$ absorption. On the contrary, in the visible spectral range the absorption band of the oxygen collision complex is not fully covered leading to a less stable retrieval as can be seen from Fig. A1 in the appendix. Additionally, strong water vapour absorption features lead to higher residuals. Altogether, this demonstrates the challenges of LP-DOAS data analysis in the visible spectral range for which reason the following study focuses on the UV retrieval. Assuring good data quality, fits are filtered for a root-mean-squares (RMS) of the residual of less than $2 \times 10^{-4}$.

During the analysis, it was found that modifications of the applied binomial high-pass filter (HP) can influence the retrieved $O_4$ column densities which was thus studied more carefully. The variation of other fit settings, e.g., the choice of fitted cross-

sections or in-/excluding an atmospheric background spectrum in the fit, has small impact compared to the choice of the high-pass filter.

Systematic differences of the measured spectra, e.g., arising from broad-band spectral variations caused by the reflectivity of the short-cut plate, require the application of a high-pass filter since the correction of the spectral features arising from these differences is hardly possible by the usage of a polynomial only (Pöhler, 2010). Also, the cut-off frequency between narrow- and broad-band absorptions is well-defined by the width of the binomial kernel of the high-pass filter. Note that the filter is defined inversely, i.e. the smaller the width the stronger the filtering effect. For instance, the filters' full width at half maximum correspond to 2.5 nm, 3.5 nm and 4.3 nm for the high-pass filters applying 4000, 8000 (standard) and 12000 iterations, respectively, given the spectral dispersion of 68.7 nm to 2048 channels. The variation of the cut-off frequency leads to a relative difference in the retrieved $O_4$ column densities well below 5% and mainly an enhanced scatter of the values in case of a too strong filter (HP 4000) as can be seen from Fig. 5. The exact choice of the high-pass filter settings has consequently no discernible impact on the results of the comparison between the measured and calculated $O_4$ concentrations which is introduced and discussed in the following.

**Table 1.** DOAS fit settings for the "standard" analysis in the UV spectral range. Different analyses will be shown in the following, including one of the $O_4$ absorption cross-sections listed in the table.

| | |
|---|---|
| Fit range | 352 – 387 nm |
| Polynomial | 3 |
| High-pass filter | 8000 |
| Cross-sections | BrO (228 K; Wilmouth et al., 1999) |
| | $NO_2$ (294 K; Vandaele et al., 1998) |
| | $O_3$ (243 K; Serdyuchenko et al., 2014) |
| | $O_4$ (various temperatures; |
| | Thalman and Volkamer, 2013; |
| | Finkenzeller and Volkamer, 2022) |
| | Atmospheric background |
| Shift & stretch | Applied to spectrum wavelengths |

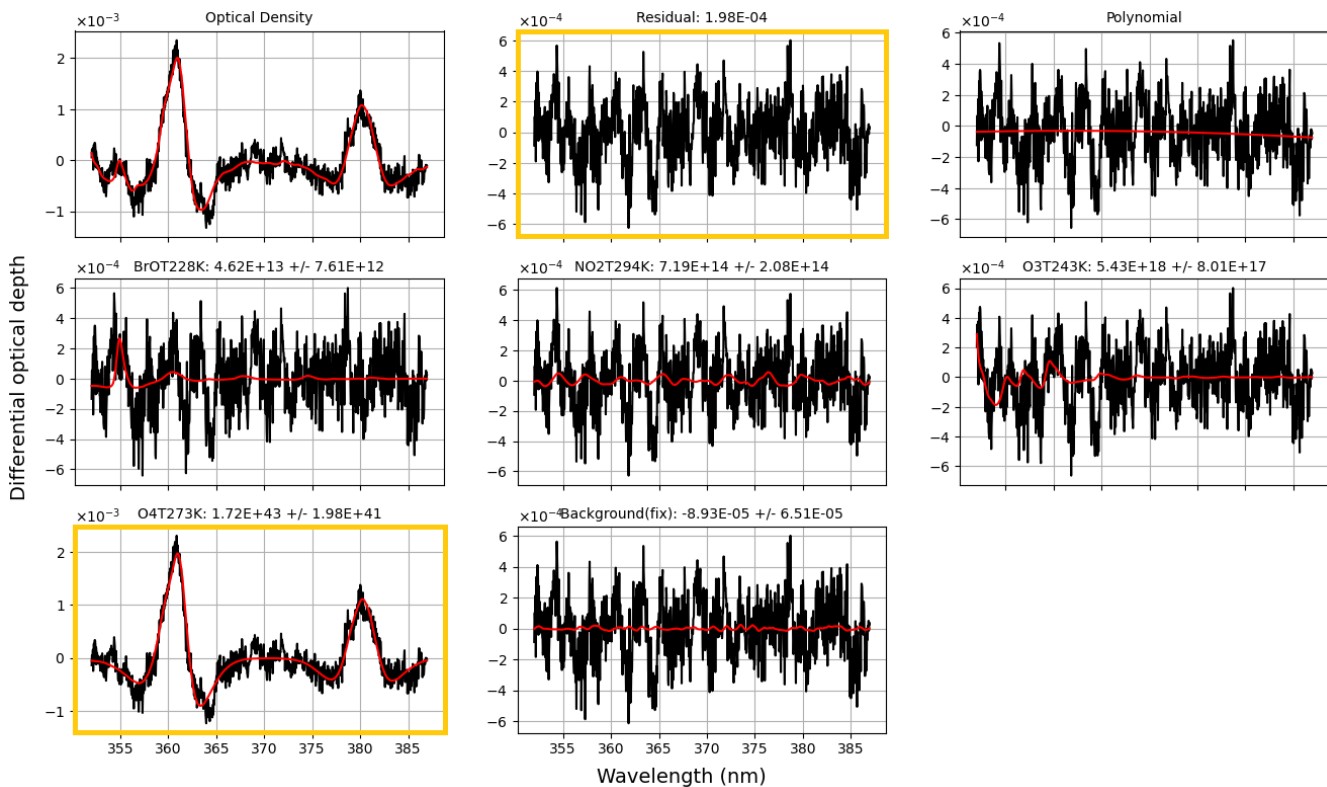

**Figure 4.** Example fit in the UV for the standard fit settings as given in Table 1, using the $O_4$ absorption cross-section at 273 K from Thalman and Volkamer (2013). The optical density panel depicts the logarithm of the high-pass filtered ratio of measured atmospheric spectrum and its respective reference, i.e., short-cut spectrum. The residual as well as the fitted $O_4$ are highlighted by the yellow frames. The remaining panels show the other fitted cross-sections as given in the title (red) and the fit plus residuum (black). The titles additionally name the temperature at which each absorption cross-section was measured and the retrieved column density in molec cm$^{-2}$ (in the case of $O_4$: molec$^2$ cm$^{-5}$).

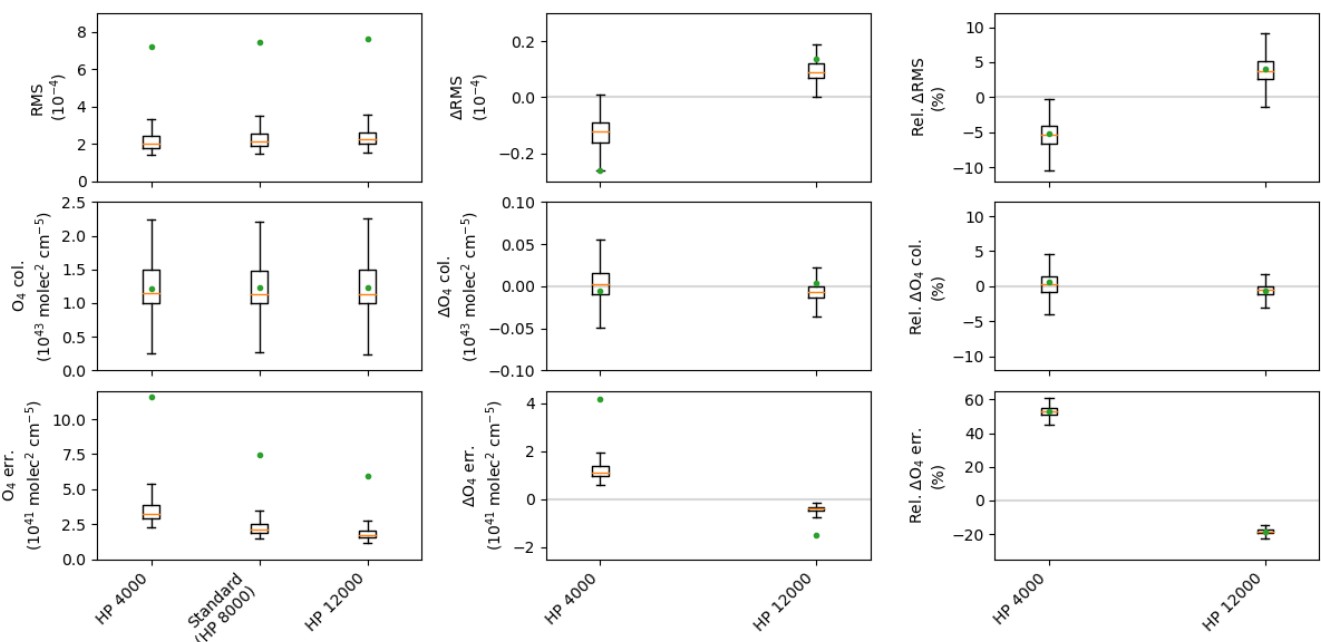

**Figure 5.** RMS, retrieved $O_4$ column and fit error in box-whisker-plots for the standard fit settings as given in Table 1, using the $O_4$ absorption cross-section at 273 K from Thalman and Volkamer (2013), as well as fit settings with a high-pass filter (HP) of 12000 and 4000 iterations as indicated in the label (left column). Absolute and relative differences to the standard retrieval are shown for all three quantities in the middle and right column, respectively. Each box extends from the lower to upper quartile values of the data (interquartile range, IQR), with an orange line at the median. The mean value is represented by a green dot. The whiskers extend to 1.5×IQR from the edges of the box, ending at the farthest data point within that interval.

## 3 Comparison between measured and calculated O$_4$ concentrations

### 3.1 Set-up of the comparison study

First, the measured O$_4$ column densities are converted into O$_4$ concentrations by dividing by the light path length which can
easily be done since the light path for LP-DOAS measurements is well-defined. Given the fact that the O$_4$ concentration is
proportional to the square of the O$_2$ concentration in the atmosphere, the expected O$_4$ concentrations can be computed via the
ideal gas law

$$pV = N k_B T$$

where $p$ is the pressure, $V$ the volume, $N$ the number of particles, $k_B$ the Boltzmann constant and $T$ the temperature. Rearranging the formula yields the O$_4$ concentration

$$c_{O_4} \propto (c_{O_2})^2 = \left( 0.21 \cdot \frac{p}{k_B T} \right)^2.$$

Since the equilibrium constant ($2 O_2 \leftrightarrow O_4$) is unknown, the unit of the O$_4$ "concentration" is molec$^2$ cm$^{-6}$. However, the
common O$_4$ absorption cross-sections are scaled accordingly such that a direct comparison to the squared O$_2$ concentration is
possible. The expected O$_4$ concentration is then calculated for each time of a spectral measurement by inserting the respective
pressure and temperature values from the meteorological long-term observations at the measurement site (Schmithüsen, 2023).
The meteorological data feature a higher temporal resolution (1 min) than the one of the LP-DOAS data (2 to 30 min), for which
reason the meteorological data was interpolated onto the time grid of the spectra for which the midst of the integration time
is reported. It should be noted that considering humidity in the above equation leads to negligible differences in the calculated
O$_4$ concentrations, which are more than an order of magnitude smaller than the observed deviations from the measured O$_4$
concentrations.

Figure 6 shows the correlation of measured and calculated O$_4$ concentrations for the complete data set covering data from
January 2016 to August 2018 and a temperature range of more than 35 K. In total, more than 69000 spectra were analysed.
The DOAS analysis was carried out with the fit settings as given in Table 1 in the UV spectral range using the Thalman and
Volkamer (2013) absorption cross-section at 253 K. Assuming that best agreement is found for an ambient temperature of the
O$_4$ absorption cross-section, i.e., in this case at 253 K, a range of O$_4$ concentrations can be calculated for this fix temperature
and considering the highest and lowest pressure values observed during the measurement period. For this range, best agreement
between measured and calculated values is expected as indicated by the grey bar. However, mostly slight underestimations of
the measured O$_4$ concentrations compared to the computed ones are observed within the shaded area. Generally, the retrieval
yields too low O$_4$ values at higher temperatures and too high O$_4$ values at lower temperatures indicating the importance of the
temperature dependence of the O$_4$ absorption.

To investigate the temperature dependence of the cross-sections, analyses are done not only with the Thalman and Volkamer
(2013) absorption cross-section at 253 K but also other available temperatures as well as the newer absorption cross-sections by
Finkenzeller and Volkamer (2022). The corresponding results are summarised in Fig. A2 and A3 in the appendix, respectively.

It can be seen that the usage of the Thalman and Volkamer (2013) absorption cross-section at 233 K yields too low $O_4$ concentrations even for temperatures around 233 K (compare grey bar, upper left plot in Fig. A2). For the absorption cross-sections at 273 K and 293 K, good agreement for measurements at the respective temperatures is found. Overall, the $O_4$ absorption cross-sections by Thalman and Volkamer (2013) show a non-linear temperature dependence with larger deviations for low-temperature cross-sections.

This finding is in accordance to the change of the spectral bands' shapes or in other words the decrease of the peak values of the $O_4$ absorption cross-sections with temperature, which are particularly important to DOAS observations, while the integrated absorption cross-sections remain independent of temperature in the case of the Thalman and Volkamer (2013) absorption cross-sections. It should, however, be noted that Finkenzeller and Volkamer (2022) found a temperature sensitivity of the integral cross-sections given the different approach to derive the spectrum baseline which is in line with stronger induced dipoles and thus greater absorption.

Nonetheless, similar results are obtained for the analyses including Finkenzeller and Volkamer (2022) absorption cross-sections shown in Fig. A3. Here, the temperature dependence of the cross-sections seems to be weaker and especially at low temperatures a better agreement to the calculated $O_4$ concentrations is observed.

Basically identical results are found for both $O_4$ absorption cross-sections at 293 K from Thalman and Volkamer (2013) and from Finkenzeller and Volkamer (2022) (compare lowermost panels in Fig. A2 and A3), respectively.

All in all, these results indicate a good agreement between measured and calculated $O_4$ concentrations for LP-DOAS measurements in the UV spectral range. Still some deviations at ambient temperatures other than those corresponding to the temperature of the used absorption cross-section are found. To eliminate these, an interpolated $O_4$ absorption cross-section could be used as will be detailed in the next section.

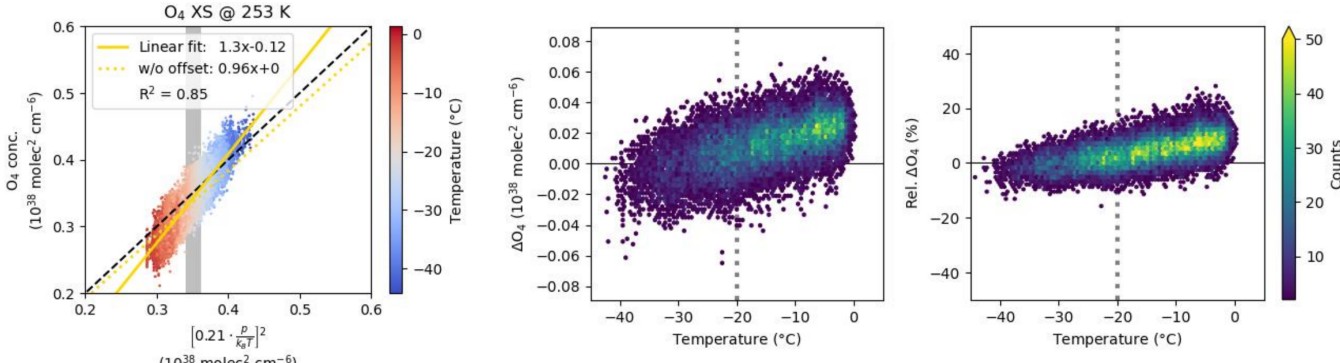

**Figure 6.** Correlation of measured $O_4$ concentrations using a fixed absorption cross-section (from Thalman and Volkamer, 2013, at 253 K, corresponding to -20°C) (ordinate) to $O_4$ concentrations calculated from ambient temperature and pressure (abscissa). Details on the retrieval and calculation are given in the main text. The grey bar indicates the range where for the chosen cross-section at 253 K best agreement between measured and calculated values is expected. Fit parameters of a linear fit with and without (w/o) intercept as well as the correlation coefficient are given in the legend. Data points are colour-coded for the respective temperature during the measurement. The middle and right panels show the absolute and relative difference between the measured and calculated $O_4$ concentration with regard to temperature, respectively. The dashed vertical line indicates the temperature of the used $O_4$ absorption cross-section.

## 3.2 Usage of interpolated $O_4$ absorption cross-sections

The advantage of LP-DOAS observations with collocated temperature measurements is the exact knowledge of the ambient temperature for each spectrum acquisition. Thus, the best matching $O_4$ absorption cross-section can be chosen individually for each spectrum during the DOAS retrieval. However, the absorption cross-sections are only available for a few specific temperatures (namely at 203, 233, 253, 273 and 293 K for Thalman and Volkamer, 2013; 223, 263 and 293 K for Finkenzeller and Volkamer, 2022). In order to match a given temperature, the set of available cross-sections is linearly interpolated to fit the required temperature. This newly calculated interpolated $O_4$ absorption cross-section is then used in the DOAS fit. Note that the results remain the same for quadratically interpolated cross-sections.

This approach differs from other studies where two absorption cross-sections of the same trace gas species but at different temperatures are included in the DOAS fit to compensate for temperature dependencies (e.g., Spinei et al., 2015).

Using these interpolated $O_4$ absorption cross-sections leads to a clear improvement in the comparison between measured and computed $O_4$ concentrations for the Thalman and Volkamer (2013) as well as Finkenzeller and Volkamer (2022) cross-sections. The results are shown in Fig. 7 and 8, respectively. As already noted, largest discrepancies exist for the Thalman and Volkamer (2013) version at low ambient temperatures (below ca. -25°C) indicating that the peak values of the $O_4$ absorption cross-section at low temperatures are too large. On the contrary, the interpolated version of Finkenzeller and Volkamer (2022) cross-sections shows nearly perfect agreement for the entire temperature range with a median difference of only $0.006 \times 10^{38}$ molec$^2$ cm$^{-5}$ (or in relative terms 2 %) and a slope of the data close to 1 (compare golden lines in Fig. 8).

This demonstrates the consistency between the $O_4$ absorption cross-sections from Finkenzeller and Volkamer (2022) measured under laboratory conditions and real atmospheric measurements.

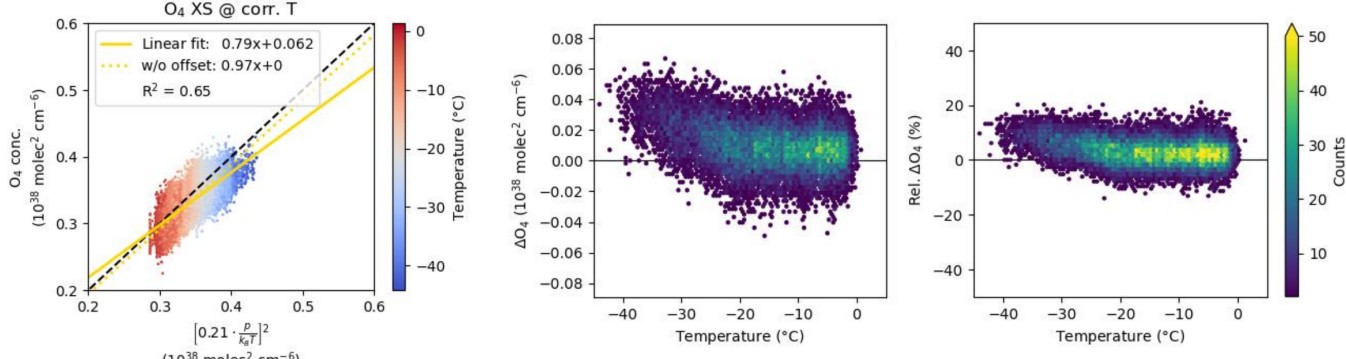

**Figure 7.** Correlation of measured $O_4$ concentrations using an interpolated absorption cross-section based on all Thalman and Volkamer (2013) cross-sections to calculated $O_4$ concentrations. Details on the retrieval and calculation are given in the main text. For the plot description, see Fig. 6.

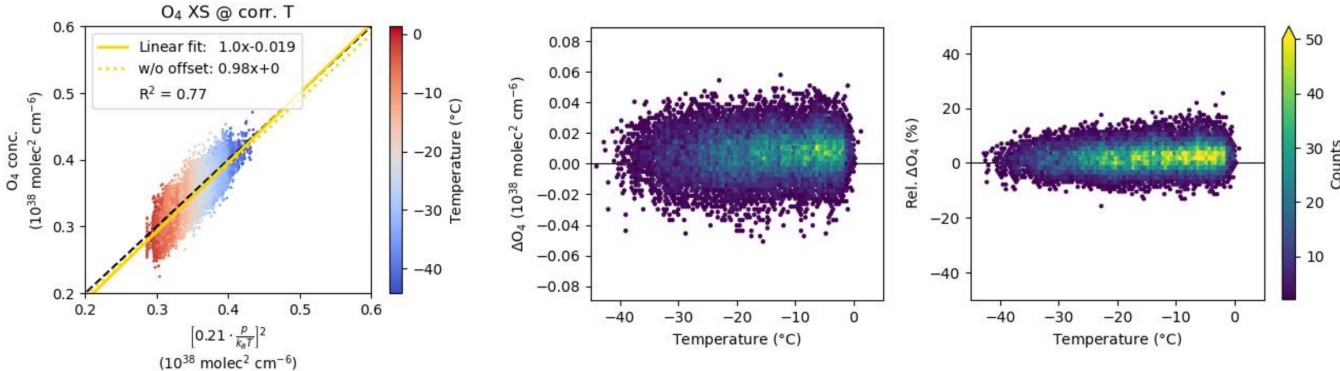

**Figure 8.** Correlation of measured $O_4$ concentrations using an interpolated absorption cross-section based on all Finkenzeller and Volkamer (2022) cross-sections to calculated $O_4$ concentrations. Details on the retrieval and calculation are given in the main text. For the plot description, see Fig. 6.

## 4 Conclusions and outlook

The comparison of measured and calculated $O_4$ concentrations shows a good agreement of the $O_4$ absorption from atmospheric LP-DOAS observations with the commonly used $O_4$ absorption cross-sections measured under laboratory conditions when properly accounting for the temperature dependence of the $O_4$ absorption. Thereby, the DOAS analysis was run for the spectral range from 352 to 387 nm covering the strong absorption band at 360 nm. Best agreement was found for the recently published $O_4$ absorption cross-sections of Finkenzeller and Volkamer (2022) and the approach of an interpolated $O_4$ absorption cross-section fitting to the ambient temperature of the measurements. The retrieval using the Thalman and Volkamer (2013) absorption cross-sections shows larger than calculated $O_4$ concentrations at low temperatures (below ca. -25°C). Overall, deviations between the measured and calculated $O_4$ concentrations are well below 20 % (mean: 7–9 %, median: 2–3 %) and indicate little to no bias (apart from the Thalman and Volkamer, 2013, cross-sections at cold temperatures).

Therefore, the small discrepancies from perfect agreement seen is this study are too small to explain the issues seen in MAX-DOAS studies. Moreover, the differences of measured and calculated $O_4$ concentrations at cold temperatures have the opposite sign to the effects that lead to the $O_4$ scaling factor. Deviations of measured and simulated $O_4$ absorptions were mainly reported in the UV, but also in the visible spectral range (see, e.g., references in Wagner et al., 2019). Hence, similar LP-DOAS measurements in the visible spectral range are needed to test the $O_4$ absorption cross-sections under ambient conditions.

This study shows that the need for a scaling factor in MAX-DOAS profile inversions is not caused by a possible systematic error of the available $O_4$ absorption cross-sections, but leaves the problem of the $O_4$ scaling factor yet unresolved. Many other hypotheses have already been tested, e.g., in Wagner et al. (2019) and Wagner et al. (2021). These studies included the investigation of the temperature dependence of the $O_4$ absorption and temperature variations along the light path of MAX-DOAS measurements, but concluded that this alone cannot explain the observed discrepancies.

Therefore, further work is needed to understand why an $O_4$ scaling factor becomes necessary in some MAX-DOAS retrievals, while this is not the case for LP-DOAS data. It should include a direct comparison of LP-DOAS and MAX-DOAS observations at the Neumayer III station, which was, however, beyond the scope of this study. Also, additional LP-DOAS measurements covering the 477 nm absorption band as well as an improved retrieval for spectra in the visible spectral range could help to enhance the understanding of the $O_4$ scaling factor. Lastly, it should be considered to repeat some of the previous studies with the newer $O_4$ absorption cross-sections of Finkenzeller and Volkamer (2022).

*Code and data availability.* Long-path DOAS data and analysis software are available upon request from the corresponding author. The auxiliary data are freely accessible online (see references in main text).

## Appendix A: Additional tables and figures

**Table A1.** DOAS fit settings for the analysis in the visible spectral range.

| | |
|---|---|
| Fit range | 550 – 585 nm |
| Polynomial | 3 |
| High-pass filter | none |
| Cross-sections | $H_2O$ (293 K; Lampel et al., 2015) |
| | $NO_2$ (294 K; Vandaele et al., 1998) |
| | $O_3$ (243 K; Serdyuchenko et al., 2014) |
| | $O_4$ (various temperatures; |
| | Thalman and Volkamer, 2013) |
| | Atmospheric background |
| Shift & stretch | Applied to spectrum wavelengths |

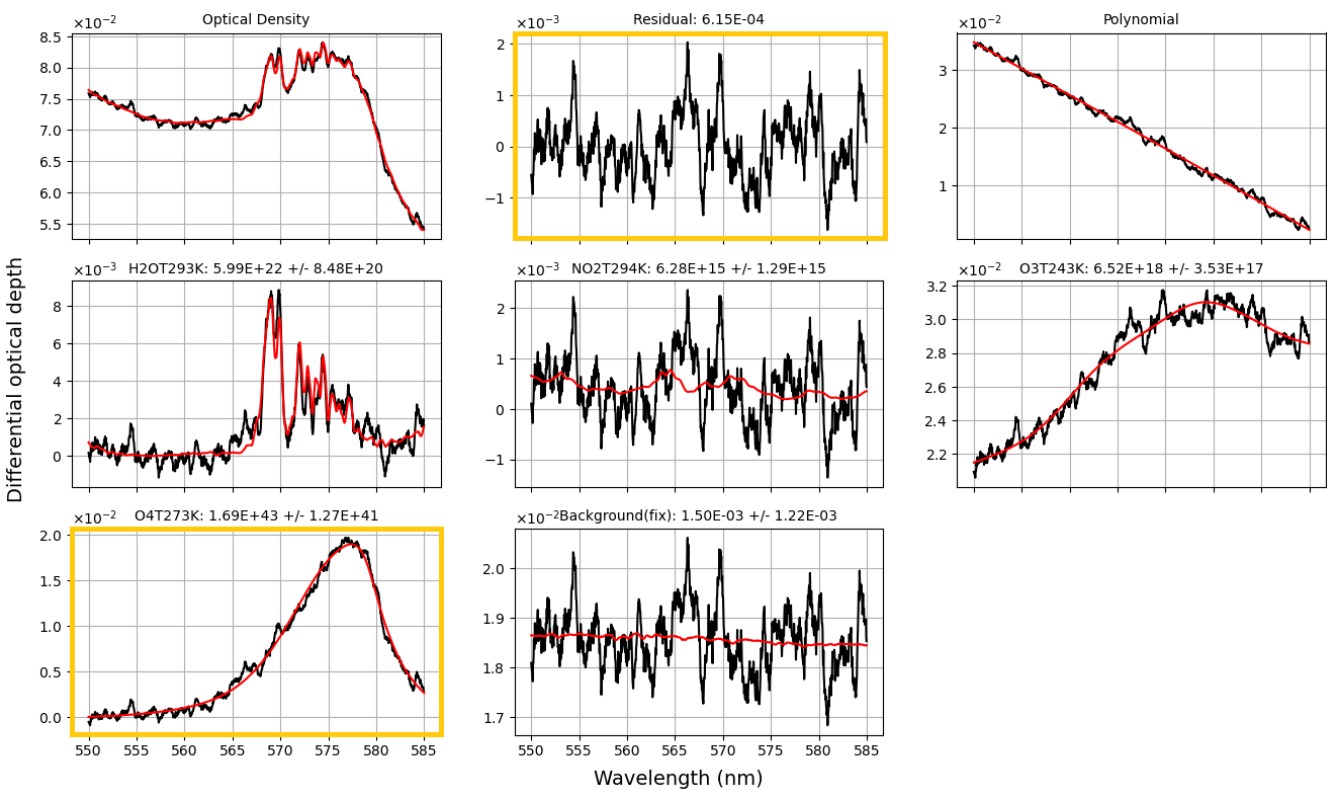

**Figure A1.** Example fit in the visible spectral range using the fit settings given in Table A1. For the plot description, see Fig. 4.

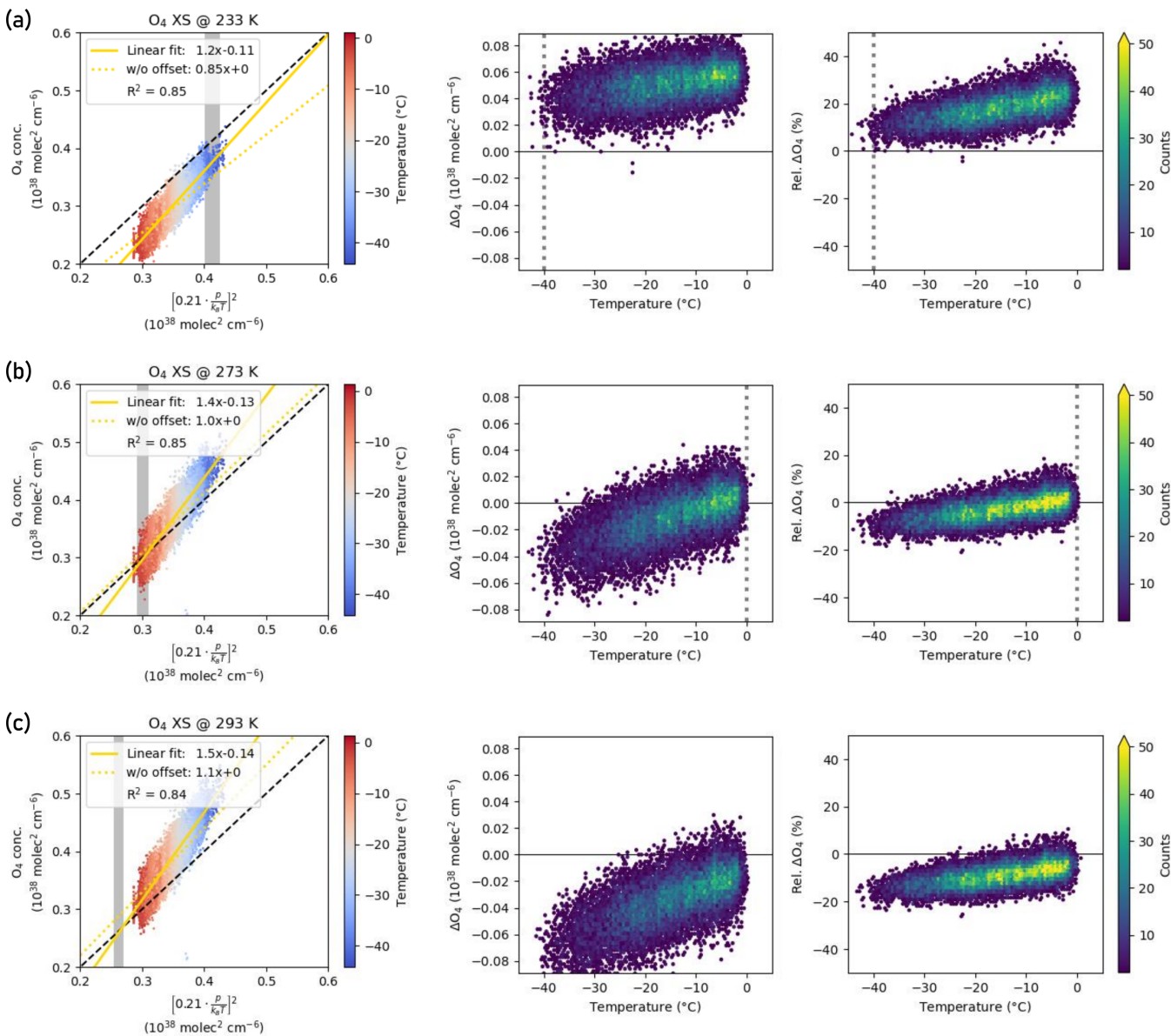

**Figure A2.** Same as Fig. 6 but for the Thalman and Volkamer (2013) absorption cross-sections at 233 K (a), 273 K (b) and 293 K (c).

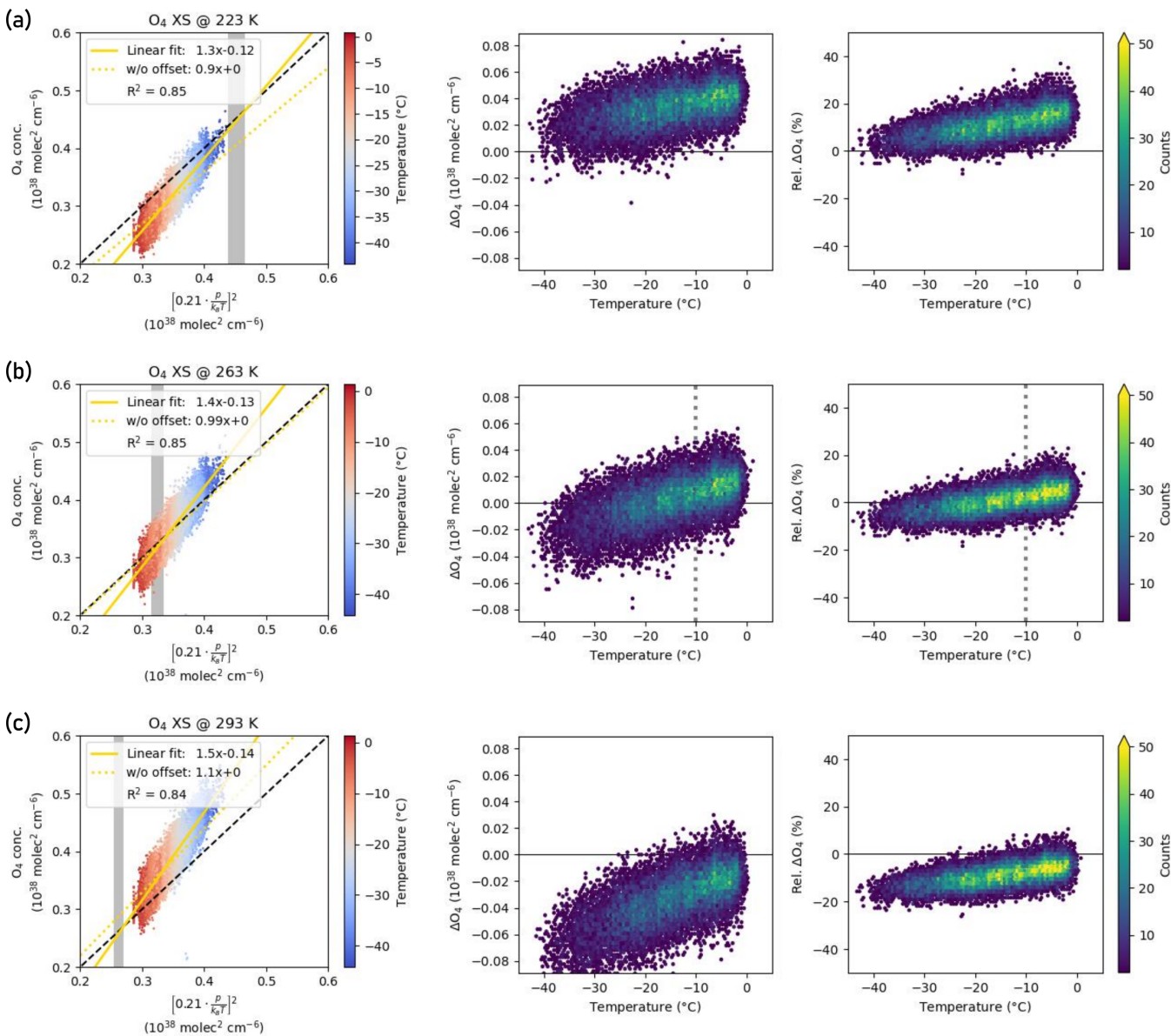

**Figure A3.** Same as Fig. 6 but for the Finkenzeller and Volkamer (2022) absorption cross-sections at 223 K (a), 263 K (b) and 293 K (c).

*Author contributions.* BL and TW designed the study. BL run the data analysis with support by UF and interpreted the results together with TW, UP and UF. JN and UF designed the LP-DOAS instrument and performed the measurements in Antarctica. UF developed the heiDOAS analysis software. BL prepared the manuscript with contributions from all co-authors.

*Competing interests.* At least one of the (co-)authors is a member of the editorial board of *Atmospheric Measurement Techniques*.

*Acknowledgements.* We acknowledge the workshop of IUP Heidelberg and all involved people for the thorough construction of the LP-DOAS instrument. Special thanks go to Denis Pöhler and Stefan Schmitt who were involved in the planning, design and set-up of the LP-DOAS instrument. Furthermore, we thank Dr. Rolf Weller and colleagues of the Alfred Wegener Institute that made it possible to acquire this data set at the Antarctic research station Neumayer III. In particular, we acknowledge the air chemists Thomas Schaefer, Zsòfia Jurànyi

and Helene Hoffmann as well as the entire 36th, 37th and 38th wintering crews that stayed at the Neumayer station and kept the LP-DOAS running throughout all weather conditions.

*Financial support.* Measurements at Neumayer III have been supported by the Deutsche Forschungsgesellschaft (DFG) in the framework of the project HALOPOLE III (grant no. FR 2497/3-2).

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
