# Peer review of "Assessment of laboratory O4 absorption cross-sections at 360 nm using atmospheric long-path DOAS observations"

_EGUsphere, 2024_

## Author Comment (AC1)

**Reply to comments from Referee #1 Henning Finkenzeller**

We would like to thank Henning Finkenzeller for the very positive and constructive comments which are addressed individually in the response below. The reviewer's comments are included in italics with the responses in blue.

*Lauster and colleagues present a study that investigates the accuracy of two O2-O2 CIA cross sections with long-path DOAS measurements. The measurements are carried out at an arctic site that experiences large temperature variations and is otherwise clean. Calculated oxygen abundances are compared to retrieved O2-O2 CIA, with varied fit settings. The study confirms that the cross sections are overall accurate and not the cause for scaling factors frequently needed in MAX-DOAS. The more recent Finkenzeller & Volkamer cross section seems to produce more accurate results for values of O2-O2 CIA, especially regarding the temperature sensitivity.*

*This scope of this study is nicely set up. The data set lends itself to this type of analysis. The logic of the analysis is appropriate. The figures are adequate. Overall, I am convinced by the work and the results outlined. I enjoyed reading the manuscript and I am looking forward to seeing this manuscript published. However, I have a couple points where I would ask the authors to refine the manuscript.*

*I do believe that referring to O2-O2 collision-induced absorption as "collision complex" or "collision pair" is wrong, and "O4" is misleading. It creates a wrong conception of the physical effect. I understand that "O4" is rooted in the MAX-DOAS community, and "O2O2 CIA" doesn't come off the lips as easily. However, I do ask the authors, particularly in light that this is a manuscript investigating our understanding of this effect, to be accurate when referring to it. Please also consider referring to the excellent monograph on collision-induced absorption by L. Frommhold. I understand that getting away from "O4" is an uphill battle, but I invite the authors to pick up this fight, and I invite the editor and other reviewers to chime in.*

Many thanks for this suggestion. We fully understand the reasoning. Nevertheless, as the reviewer correctly mentioned: '"$O_4$" is rooted in the MAX-DOAS community. Therefore, we will stick to this established notation, but we also mention in the abstract and introduction the correct notation and explain the reason for still using "$O_4$".

We have hereto adapted the beginning of the abstract from:

"The atmospheric absorption of the oxygen collision complex $O_2$-$O_2$, [...]"

to:

"Absorption of light in the atmosphere by collision-induced absorption by two oxygen molecules $O_2$-$O_2$, [...]"

and also changed the introduction from:

"Atmospheric absorption of the oxygen collision complex $O_2$- $O_2$, [...]"

to:

"Absorption of light in the atmosphere by the oxygen collision complex $O_2$-$O_2$, [...] The collision of two oxygen molecules gives rise to the formation of several absorption bands through so-called collision induced absorption (CIA). CIA describes the absorption of systems of interacting atoms or molecules where the absorption exceeds the simple sum of the absorption by the isolated atoms or molecules (Frommhold, 2006; Finkenzeller and Volkamer, 2022, and references therein). Since CIA does not involve a bound state, the term $O_4$ might be misleading. However, this term is rooted in the MAX-DOAS community for which reason the notation $O_4$ will be used in the following to denote $O_2$-$O_2$ CIA.

*17: what is "atmospheric" in "atmospheric absorption"*

Please see comment above.

*18: See main comment. Maybe something along the lines of "O2-O2 CIA occurs proportionally to the square of the oxygen concentration." could work.*

We have rephrased the sentence:

"The absorption of $O_4$ occurs proportionally to the square of the $O_2$ concentration."

*37: What are the differences and common aspects of the two cross sections? Please help the reader to understand the difference.*

We have added the following paragraph for better understanding the differences and similarities of both sets of absorption cross-sections:

"While both Thalman and Volkamer (2013) and Finkenzeller and Volkamer (2022) use a similar experimental set-up of a cavity enhanced (CE-) DOAS instrument, there are some improvements implemented for the more recent study optimising the optical components and thus the stability of the measurements. Moreover, pure oxygen was used instead of air mixtures to enhance the signal-to-noise considerably and allowing to capture also weak absorption features. Both studies report that the peak cross-section increases and the band width decreases at colder temperatures. However, Finkenzeller and Volkamer (2022) observe an increase of the integral cross-section with temperature which was not seen by Thalman and Volkamer (2013). This deviating observation is likely due to the assumption that the absorption cross-section is zero in the minima between neighbouring absorption bands to approximate the baseline of the absorption cross-section in Thalman and Volkamer (2013) which is not applied in Finkenzeller and Volkamer (2022)."

*90: What is the resolution of the cross sections? What is the resolution of the spectrometer? (I think this is relevant information that should be within this manuscript.)*

We have added the following paragraph:

"The spectral resolution of the $O_4$ absorption cross-sections ranges from 0.32 to 0.45 nm and 0.31 to 0.42 nm for Thalman and Volkamer (2013) and Finkenzeller and Volkamer (2022), respectively. Since the observed absorption features are 10 to 30 times wider and also the spectral resolution of the LP-DOAS (0.54 nm) is lower, the absorption cross-sections can be considered as ``fully resolved'' and are directly convolved with the respective instrument slit function."

The spectral resolution of the LP-DOAS instrument is about 0.54 nm and is added to the description of the instrument properties in Sect. 2.2.

*97-99: Is this novel? If not, please reference the method.*

We added a reference to Pöhler (2010) who also used this method and describes the effect of an incomplete background correction.

*110-118: I was left a little unsatisfied about why 8000 iterations were identified as best setting. How would the further interpretation change if another value was chosen? How does the value affect the seemingly persistent offsets in the O2-O2 CIA? This should be discussed further. Maybe a duplicate of Figure 8, but with another iteration number, could be interesting.*

The interpretation would not at all change even if different settings for the high-pass filter were chosen. As mentioned in the text, a too strong filtering results in a larger scatter of the retrieved $O_4$ concentrations. However, the values change by far less than 5% which does not impact the further comparisons to the expected, calculated $O_4$ concentrations. Also, no bias is introduced by the exact choice of the filter (see

Fig. 5). Nonetheless, we ran additional analyses with different high-pass filter settings but using the interpolated $O_4$ absorption cross-sections which confirm the findings above.

[Figure]

Same as Fig. 7 (Thalman and Volkamer, 2013) but using a stronger (HP 4000) or weaker (HP 12000) high-pass filter than the original version (HP 8000) on the left and right side, respectively.

Apart from slightly different correlation coefficients, the overall result remains the same as in the paper.

[Figure]

Same as Fig. 8 (Finkenzeller and Volkamer, 2022) but using a stronger (HP 4000) or weaker (HP 12000) high-pass filter than the original version (HP 8000) on the left and right side, respectively.

Again, no obvious difference between the versions with stronger/weaker filtering can be seen.

We have added the following sentence to the end of Sect. 2.3:

"The exact choice of the high-pass filter settings has consequently no discernible impact on the results of the comparison between the measured and calculated $O_4$ concentrations which is introduced and discussed in the following."

*Table 1: The number of iterations for the high-pass filter should be 8000, rather than 4000, shouldn't it?*

Many thanks for pointing this out! The value in the table is now corrected (HP 8000 as stated in the text).

*Figure 4: The highlighting frame does not work well. Could be left out or improved.*

We have changed the highlighting frame to yellow.

*Figure 5: Which cross section underlies these data? Currently not clear. The link to the table is broken in the caption.*

Thank you. The link is working now and we have added the information regarding the used absorption cross-section (Thalman and Volkamer, 2013, at 273 K) in the caption of Fig. 4 and 5.

*128: How does humidity play in here? In the arctic it won't matter much, but it should be discussed as being generally important. Is 21% sufficiently accurate? I believe 20.95% might be more accurate (0.1% difference in O2-O2 CIA).*

Calculating the $O_4$ concentration with 20.95% instead of 21% will result in a change of 0.0014 to 0.0021e38 $molec^2$ $cm^{-6}$ which is more than one order of magnitude smaller than the deviation between measured and calculated $O_4$ concentrations up to 0.04e38 $molec^2$ $cm^{-6}$ even for the "best matching" analysis using the interpolated absorption cross-section based on Finkenzeller and Volkamer (2022) (compare middle panel of Fig. 8).

We added the following sentence to the manuscript:

"It should be noted that considering humidity in the above equation leads to negligible differences in the calculated $O_4$ concentrations, which are more than an order of magnitude smaller than the observed deviations from the measured $O_4$ concentrations."

*137: Do I understand correctly that the pressure changes were not considered? A pressure change of 2% (1000 vs 980 hPa) would lead to a O2-O2 CIA change of 4%, rather substantial. Accounting for the pressure could improve the closure – or make it worse… Please consider considering the pressure, or discuss the effects of not considering it a little more.*

The ambient pressure is considered in the calculation of the expected $O_4$ concentrations according to the given equation in Sect. 3.1. The grey bar in the figures indicates the range for which best agreement is expected. This corresponds to the range of values which the $O_4$ concentrations can have if using a fixed temperature (corresponding to the one of the used absorption cross-section for analysing the spectra) and maximum/minimum pressure (over the complete observation period).

We expanded the sentence for better readability:

"Assuming that best agreement is found for an ambient temperature of the $O_4$ absorption cross-section, i.e., in this case at 253 K, a range of $O_4$ concentrations can be calculated for this fix temperature and considering the highest and lowest pressure values observed during the measurement period. For this range, best agreement between measured and calculated values is expected as indicated by the grey bar."

*142: The temperature dependence pertains not only to the peak absorption cross section, but also shape and integral cross section. Better to eliminate the parenthesis "i.e., the strength of its peak value".*

Done.

*145: The reference to Fig A2 is misplaced, it should only be a reference to A3?!*

Here we refer to both the results using the Thalman and Volkamer (2013) absorption cross-sections at other temperatures than the example in the main part of the paper (Fig. A2) and the results using the Finkenzeller and Volkamer (2022) absorption cross-sections (Fig. A3).

*146,147: Temperature should be 223 K, not 233 K?!*

Thank you. There was a mistake in the caption of Fig. A2 which is now updated. The given temperatures in the text are correct.

*153: I disagree with the statement that the integral cross sections are independent of temperature. Thalman & Volkamer were not on point in this regard, but presumably rather intended to argue that there is no bound state ("O4"). The more recent Finkenzeller & Volkamer study found a temperature sensitivity, which is also expected from physics: "With increasing temperature closer encounters occur, which leads to stronger induced dipole moments and thus greater intensities [1]."*

We have adapted the following paragraph:

"This finding is in accordance to the change of the spectral bands' shapes or in other words the decrease of the peak values of the $O_4$ absorption cross-sections with temperature, which are particularly important to DOAS observations, while the integrated absorption cross-sections are independent of temperatures (Thalman and Volkamer, 2013)."

adding information on the more recent findings by Finkenzeller and Volkamer (2022):

"[…] while the integrated absorption cross-sections remain independent of temperature in the case of the Thalman and Volkamer (2013) absorption cross-sections. It should, however, be noted that Finkenzeller and Volkamer (2022) found a temperature sensitivity of the integral cross-sections given the different approach to derive the spectrum baseline which is in line with stronger induced dipoles and thus greater absorption."

*168: The interpolation of temperatures has been used previously. Please add (a) reference(s).*

To compensate the temperature dependence of absorption cross-sections, two absorption cross-sections of the same trace gas species but at different temperatures can be included in the DOAS fit. This is, for example, used in Spinei et al. (2015) investigating temperature and pressure dependencies of the collision-induced absorption of oxygen. However, in the here-presented study, the temperature interpolation is done before the DOAS fit and only one explicit $O_4$ absorption cross-section is then used.

We added the reference to the manuscript.

*Figure A2: I was misled by the caption. It was not clear to me that the temperatures are the interesting parameter, rather than the origin of the O2-O2 CIA cross section. I suggest starting the caption with something like "Sensitivity of O2-O2 CIA to temperature. …"*

We kept the caption since both the temperatures and the origin of the cross-section are equally important.

---

## Author Comment (AC2)

**Reply to comments from Referee #2**

We would like to thank the referee for the very positive and constructive comments which are addressed individually in the response below. The reviewer's comments are included in italics with the responses in blue.

*This is an important study that aims to investigate the inconsistencies in MAX-DOAS measurements of O2-O2 collision induced absorption (CIA) and radiative transfer models. To this end, the authors analyzed two years of observations from a (LP-) DOAS instrument deployed at the Antarctica station. In particular, they investigated the relation between measured and modeled O2-O2 (CIA) over a wide range of temperatures. Especially interesting is their access to the lower temperatures as a point of controversy or lack of data from experimental measurements. They have concluded that the set of laboratory data recently published by Finkenzeller and Volkamer provides the best agreement, however the discrepancies in MAX-DOAS are not completely understood. It is an important paper using field data to evaluate reference spectroscopic information and should be published after following comments are addressed:*

*Major comment:*

*One should not expect constant or even linear dependence of CIA on temperature, whether looking at integrated band absorption or at individual spectral points. In particular for integrated absorption one should expect that the CIA should be more intense at low T because of bound states, and more intense at high T because it's easier for colliding molecules to get close together, and in between there must be some minimum. See, for instance, a paper by Vigasin on the temperature variation of intensity of the CIA underlying oxygen fundamental (https://doi.org/10.1016/j.jms.2004.02.003). In the Figure 1 of that paper you will see that the temperature dependence is not expected to be constant or linear.*

*Also, the distribution of the intensity within a band also should vary with temperature because of the Boltzmann population of energy levels.*

*Could the interpolation potentially be done as a quadratic function combining the data from 2013 and 2022 papers from Volkamer's group? The earlier work can be given enhanced uncertainties but still could be useful. In any case the 2013 data should not be fit to linear function. It would be good to include the figures associated with interpolation*

We ran additional analyses testing the quadratic interpolation as suggested. We find no discernible difference in the results if considering the Thalman and Volkamer (2013) or the Finkenzeller and Volkamer (2022) absorption cross-sections separately as done in the paper draft using linear interpolation (compare figures below and in the paper draft).

[Figure]

Same as Fig. 7 but for the case of quadratic interpolation of the Thalman and Volkamer (2013) absorption cross-sections.

[Figure]

Same as Fig. 8 but for the case of quadratic interpolation of the Finkenzeller and Volkamer (2022) absorption cross-section.

A combination of both sets of absorption cross-sections by Thalman and Volkamer (2013) and Finkenzeller and Volkamer (2022) leads to a worsening of the correlation (see below). This is probably caused by the strong deviations from the calculated $O_4$ concentrations at low temperatures when considering the Thalman and Volkamer (2013) version. Especially, since the Finkenzeller and Volkamer (2022) absorption cross-sections alone yield very good results.

Furthermore, it should be noted that the temperatures considered in this work (223 to 293 K) do not show a strong quadratic relation also in the study by Vigasin (2004). A large difference between linear or quadratic interpolation would therefore not be expected.

[Figure]

Same as Fig. 7 and 8 but for the case of quadratic interpolation considering both Thalman and Volkamer (2013) and Finkenzeller and Volkamer (2022) absorption cross-sections.

We have added a note in the manuscript that quadratic interpolation of the given absorption cross-sections does not change the results.

**A couple of minor suggestions**

*You can mention that Thalman and Volkamer CIA is currently recommended by HITRAN*

The HITRAN recommendation was added to the introduction (including its respective publications: Karman et al., 2019 and Gordon et al., 2022).

*In line 166 change "…a couple.." to "..a few…"*

Done.

---

## Author Comment (AC3)

**Reply to comments from Referee #3**

We would like to thank the referee for the very positive and constructive comments which are addressed individually in the response below. The reviewer's comments are included in italics with the responses in blue.

*The paper by Lauster et al. presents long-path DOAS measurements in the remote location of Neumayer Station 3 in Antarctica to retrieve atmospheric O4 concentrations and to assess the accuracy of the laboratory O4 absorption cross-section, commonly used in DOAS applications. In the past, some studies reported deviations between MAX-DOAS O4 measurements and results from radiative transfer simulations, while others showed none. The authors questioned if the used O4 cross section in the DOAS fit can explain these deviations. Different O4 cross-sections at different temperatures are investigated in the UV spectral range. The study provides more insights into this debate and shows that the retrieved O4 concentrations align with the expected concentrations calculated from meteorological measurements on-site. However, better agreement is found for the newer version of the commonly used O4 cross-section.*

*The study covers an ongoing debate in the DOAS community, is well-structured, and provides further insights from another point using long-path DOAS observation.*

*I recommend publication after including the following points.*

***Specific comments:***

*Line 35: I am a bit confused about the wavelength range of 352 to 387 nm given here and the grey bar given in Figure 1, which are not the same. What are these wavelength ranges, the spectral range of the spectrometer, and the fitting window, maybe you can clarify this in the text shortly.*

Thank you for this comment. The 352 to 387 nm refers to the chosen wavelength range of the DOAS fit while the grey bar in Fig. 1 represents the total wavelength range covered by the spectra. This information was added to the text:

"In this study, long-term long-path (LP-) DOAS observations are used to examine the $O_4$ absorption at 360 nm. The spectra measured in the UV spectral range cover the wavelengths from about 327 to 395 nm, however, a shorter fit window from 352 to 387 nm was chosen here. Details on the data analysis are given in the next section."

*Figure 1: The unit of the cross-section on the y-axis is missing.*

The unit was added.

*Figure 1: You mention that the 477 nm band is not covered, I think you can additionally mention that the 577 nm band is at the edge of the LP-DOAS visible wavelength range.*

The description was adapted according to your comment.

*Line 55: Why have direct sun measurements further difficulties due to "small atmospheric absorption"? What is causing this problem, is it only the light path, which is shorter in the lower troposphere compared to MAX-DOAS, is this relevant for O4 absorption?*

We have rephrased the sentence to better explain the relevant drawbacks of direct sun measurements:

"Especially at short wavelengths, direct sun measurements, despite the well-defined light path (at high elevation angles), experience further difficulties such as small atmospheric absorptions or substantial contributions of scattered sun light (at low elevation angles)."

*Line 58: MAX-DOAS and direct sun measurements don't "measure vertical column densities". I think both "measure" and "vertical" are technically not correct here, I suggest changing it to "retrieve" or/and changing it to slant column densities along the light path.*

Thank you. We have implemented your comment. This part of the sentence now reads:

"[…] in MAX-DOAS retrievals or direct sun measurements, from which the atmospheric column density is derived considering vertical profiles of temperature and pressure. […]"

*Line 75: I think it is not clear for the reader what kind of conditions are needed for your study. Large temperature range is good for analyzing the cross-sections at different temperatures. What is the low aerosol optical depth good for in your study – less attenuation? Why is this helpful?*

We have added to following to enhance the understanding why low aerosol optical depths are important:

"This allows for long light paths and the concomitant strong $O_4$ absorption offers a good signal-to-noise ratio."

*What kind of measurements are used for your study/are you interested in? Do you use all available measurements from January 2016 to August 2019, do you filter for specific measurements, do you filter out bad weather? What is the temporal resolution of the measurements?*

All available measurements are used without filtering for specific weather conditions (information added to the manuscript). To assure good data quality, DOAS fits are filtered for a root-mean-square (RMS) of the residual of less than 2e-4 which is also noted in the data analysis section. The temporal resolution of the measurements (about 2 to 30 minutes) is added to the manuscript (see next comment).

*I think the section lacks information about the LP-DOAS instrument. The spectrometer is not mentioned at all; what kind of spectrometer, which wavelength range, which spectral resolution is used?*

We have added the following information about the LP-DOAS instrument:

"Spectra are then captured by an Acton 300i spectrometer using a holographic grating (1200 gr. mm-1) with attached Andor DU440 BU CCD. This set-up allows for a spectral resolution of ca. 0.54 nm covering a spectral window of about 65 nm. The measured spectra have a temporal resolution of about 2 to 30 minutes […]"

The temporal resolution varies between successive measurements of the same spectral window (2 min) and the time period during which other spectral windows are measured (30 min). There can be larger gaps during bad weather periods or due to technical problems, however, these are not critical for the presented study.

Additionally, the type of the laser driven light source (EQ-99X) was added to the text for completeness.

*Figure 3: Monthly averages over which period, 2 years (2016-2017)?*

The time period of the monthly averages (2016-2017) was added to the caption.

*Line 86: To clarify for people having no/little knowledge about LP-DOAS, add something like "creating a short-cut for the light, which stays inside the telescope and goes directly into the spectrometer"*

Thanks for your comment. We have added "creating a short-cut for the light, which does not traverse the atmosphere but enters the spectrograph directly".

*Line 95: The high-pass filter is only needed in the UV but not in the visible (Table A1), is this right?*

This is correct. Contrary to the UV, the usage of a high-pass filter has not proved to be beneficial for the retrieval in the visible spectral range.

*Line 101-104: That the O4 bands in the visible are not covered is not a general problem but only of this specific instrument, right? If you would have a different spectrometer you could cover other wavelengths.*

This is correct. There is a movable grating turret within the spectrograph of this instrument which can be adjusted and thus different wavelength regions can be covered. Since the main purpose of the measurements were focused on halogen chemistry (Nasse, 2019) rather than the $O_4$ study presented here, the spectral windows do not cover all $O_4$ absorption bands.

*Line 115: You mention that the HP 8000 is the standard, however, in Table 1, you write 4000, please clarify.*

Many thanks for pointing this out! The value in the table is now corrected (HP 8000 as stated in the text).

*Line 131: What is the temporal resolution of the LP-DOAS and the meteorological data, how is the matching done?*

The meteorological data has a temporal resolution of 1 minute which is higher than the one of the LP-DOAS data (2-30 minutes). Thus, the meteorological data was interpolated onto the time grid of the spectra for which the midst of the integration time is reported.

This information was also added to the manuscript.

*Line 139: How is this grey bar defined?*

We have adapted the following passage:

"Assuming that best agreement is found for an ambient temperature of the $O_4$ absorption cross-section, i.e., in this case at 253 K, and taking into consideration the slight pressure differences during the measurement period, the best agreement between measured and calculated values is expected where indicated by the grey bar."

to:

"Assuming that best agreement is found for an ambient temperature of the $O_4$ absorption cross-section, i.e., in this case at 253 K, a range of $O_4$ concentrations can be calculated for this fix temperature and considering the highest and lowest pressure values observed during the measurement period. For this range, best agreement between measured and calculated values is expected as indicated by the grey bar."

*Figure 6: How is the grey bar defined? Can you add it to the middle and right plots as well? Change "values" to "O4 concentrations" in the caption.*

The grey bar takes into consideration the slight pressure differences during the measurement period. It is not possible to add this to the other plots in the same manner. Instead, a dashed vertical line was added at the temperature of the used $O_4$ absorption cross-section. The caption was adapted accordingly.

*Figure 7 and 8: Change "values" to "O4 concentrations" in the caption.*

Done.

*Line 180-186: I think you should be here more precise – provide some numbers. What is a good agreement, what is needed/good enough? How good is your agreement for the 2022/2013 cross-section retrieval with the calculated concentrations?*

We added the following sentence to the conclusions:

"Overall, deviations between the measured and calculated $O_4$ concentrations are well below 20% (mean: 7—8 %, median: 2—3 %) and indicate little to no bias (apart from the Thalman and Volkamer, 2013, cross-sections at cold temperatures)."

*Which wavelength ranges were used in the studies that are in need or don't need the scaling factor, are there deviations, are they also in the UV – like the focus of this study? How well can you draw conclusions for the visible, maybe the problem is in the visible only?*

We added the following paragraph to the conclusions:

"Deviations of measured and simulated $O_4$ absorptions were mainly reported in the UV, but also in the visible spectral range (see, e.g., references in Wagner et al., 2019). Hence, similar LP-DOAS measurements in the visible spectral range are needed to test the $O_4$ absorption cross sections under ambient conditions."

*You mention that the previous studies used the 2013 cross-section, for which you also see discrepancies with your measurements. Can you comment on this, are these discrepancies too small to explain the issues seen in these studies?*

We added the following paragraph to the conclusions:

"Therefore, the small discrepancies from perfect agreement seen is this study are too small to explain the issues seen in MAX-DOAS studies. Moreover, the differences of measured and calculated $O_4$ concentrations at cold temperatures have the opposite sign to the effects that lead to the $O_4$ scaling factor."

***Technical corrections:***

*Line 40: This sentence is hard to read; maybe something like "At first, differences between MAX-DOAS and LP-DOAS measurements are introduced to answer the question of why LP-DOAS observations are well suited to further investigate the reported inconsistencies from MAX-DOAS studies."*

Thank you. We have implemented your comment.

*Line 58: "consider vertical profiles of temperature and pressure" instead of "consider temperature and pressure vertical profiles"*

Done.

*Line 66: I was confused by the "(Met retro)" and "(Atka retro)" when reading this, without retro reflectors mentioned before at all, and haven't looked yet in Figure 2. I suggest adding something like "across a distance of 1.55 km to the closer retro reflector (Met retro), or to another retro reflector at 2.95 km (Atka retro)."*

Thank you. We have implemented your comment.

*Line 66: The sentence is hard to read, I suggest something like "Depending on the prevailing weather conditions, the amount of reflected light varies, and the light path can be chosen depending on the atmospheric conditions to optimize the amount of received light and covered light path length. "*

Thank you. We have implemented your comment.

*Line 75: Add a link to the AERONET webpage.*

The link can be found if clicking on the NASA-GSFC reference which is given in the text.

*Line 76: Change "measurement days" to "measurements"*

Done.

*Line 87: To improve readability, change "in addition by shutting off the light source" to "without the light source".*

Done.

*Line 93: Delete one "the"*

Thank you. Done.

*Line 95: To clarify, change to: "Different analyses will be shown in the following, including one of the O4 absorption cross-sections listed in the table."*

Done.

*Table 1: Change "various" to "various temperatures". Add something like this to the caption to clarify that only one cross-section is used at one temperature at a time: "Different analyses will be shown in the following, including one of the O4 absorption cross-sections listed in the table."*

Done.

*Caption Figure 5: The link to the table is not working.*

Thank you. The link is working now.

*Line 128: Use a point instead of a cross for the multiplication sign.*

Done.

*Line 134: This is more than two years, maybe just write "covering data from January 2016 to August 2018".*

Done.

*Line 139: I would argue here on the retrieved O4 concentrations instead of the calculated, which you also do in the next sentence and the following section. I think it would improve readability.*

Done.

*Line 153: Brackets missing around reference.*

Done.

*Line 154: Shown where?*

Done.

*Line 186: shows instead of show*

Done.

*Line 186: Change "expected" to "calculated".*

Done.

---

## Author Response (AR2)

**Reply to Editor decision**

We would like to thank Michel Van Roozendael for the decision to publish our manuscript after the following minor revision. The public justification by the Editor reads:

*Although the first referee insists on a change of the notation used for the O2-O2 collision-induced absorption (CIA), the same request is not made by the other referees. Given the fact that the O4 notation is commonly used in the DOAS community and that all figures in the manuscript use this notation, I support the compromise proposed by the authors, i.e. clearly explain in the introduction the exact meaning of the CIA and the motivation to use the abbreviated O4 notation in the rest of the paper. That said, and perhaps to take the compromise a step further, I'd suggest also using the exact notation (O2-O2 CIA) in the title of the manuscript. In this way, it becomes clear for the readers that the authors are well aware of the particular nature of the O2-O2 absorption bands.*

We have implemented the Editor's comment and changed the title to

"Assessment of laboratory $O_4$ ($O_2$-$O_2$ collision induced) absorption cross-sections at 360 nm using atmospheric long-path DOAS observations"

Also, the abbreviation for collision-induced absorption (CIA) was added to the first sentence of the abstract for completeness.